# The Importance of Complementary PCR Analysis in Addition to Serological Testing for the Detection of Transmission Sources of *Brucella* spp. in Greek Ruminants

**DOI:** 10.3390/vetsci9040193

**Published:** 2022-04-17

**Authors:** Anthimia Batrinou, Irini F. Strati, Andreas G. Tsantes, Joseph Papaparaskevas, Ioannis Dimou, Dimitrios Vourvidis, Anna Kyrma, Dionysis Antonopoulos, Panagiotis Halvatsiotis, Dimitra Houhoula

**Affiliations:** 1Department of Food Science and Technology, University of West Attica, 12243 Egaleo, Greece; batrinou@uniwa.gr (A.B.); estrati@uniwa.gr (I.F.S.); giannisdimou93@gmail.com (I.D.); antondion@uniwa.gr (D.A.); 2Microbiology Department, ‘Saint Savvas’ Oncology Hospital, 11522 Athens, Greece; andreas.tsantes@yahoo.com; 3Department of Microbiology, Medical School, National and Kapodistrian University of Athens, 15772 Athens, Greece; ipapapar@med.uoa.gr; 4Laboratory of Brucellosis and Bacterial Disease Serology, Department of Diagnostic Pathology, Anatomy, Histology and Microbiology Directorate of Athens Veterinary Center, Ministry of Rural Development and Food, 11522 Athens, Greece; dvourvidis@gmail.com (D.V.); annakirma@gmail.com (A.K.); 52nd Propaedeutic Department of Internal Medicine, Medical School, National and Kapodistrian University of Athens, “ATTIKON” University Hospital, Chaidari 12462, Greece; pahalv@gmail.com

**Keywords:** brucellosis, ruminants, PCR, antibodies, DNA, RBPT

## Abstract

The early and accurate diagnosis of brucellosis, a ubiquitous zoonotic infection, is significant in preventing disease transmission. This study aimed to assess the infection rate of *Brucella* spp. in ruminants and to evaluate the agreement between a serological test and a molecular method for the detection of infected cases. Blood and milk samples of 136 ruminants were analyzed using two laboratory methods: the Rose Bengal plate (RBP) test to detect *B. abortus* and *B. melitensis* antibodies and the molecular polymerase chain reaction (PCR) method for the presence of bacterial DNA. The agreement between the methods was assessed using the kappa statistic. Based on the RBP test, there were 12 (8.8%) seropositive animals (10 sheep and 2 cows), while 2 (1.4%) samples were positive on PCR analysis. The positive PCR samples were from seronegative cow samples on RBP testing. There was slight agreement (k = −0.02) between the two methods, which was not statistically significant. Our results indicate that complementary molecular methods are useful to detect the bacteria in infected animals that are seronegative due to an early stage of infection. Therefore, a combination of molecular methods and serological tests can be applied to detect brucellosis in ruminants efficiently.

## 1. Introduction

Brucellosis is an important zoonotic disease that causes significant reproductive disorders in livestock worldwide. The most commonly isolated pathogen in sheep and goats is *B. melitensis*, whereas the most commonly isolated species in humans include *B. melitensis* and *B. abortus* [1,2,3]. Brucellosis can rapidly spread among caprine and ovine herds, resulting in multiple organ dysfunction [4,5,6]. Brucellosis is also associated with a devastating economic impact on the international trade of milk, meat, and related products due to a substantial decrease in ruminant production. Several causes, such as loss of milk production, high rates of late pregnancy abortions and stillbirths due to these infections, contribute to decreased ruminant production [7,8].

The incidence of human brucellosis in Greece remains high [9]. According to the European Safety Authority Report in 2021, Greece is one of the European countries in which brucellosis is not officially eradicated, along with Italy, Portugal, Croatia and Austria. Greece reported an incidence of 0.61 positive cases per 100,000 population, 10 times higher than the average rate of confirmed cases in humans in the European Union region. Regarding the enzootic incidence of brucellosis in Greece, brucellosis was reported in 122 cattle herds in 2018 and 85 cattle herds in 2019, while 36 positive sheep and goat herds in 2018 and 37 positive sheep and goat herds in 2019 were also reported. The data on sheep and goat brucellosis in Greece derives from the national eradication program that monitors animal herds on Greek islands [10]. The relatively high infection rate in cattle herds (2.8%) and sheep and goat herds (3.3%) highlights the necessity for early detection of infected herds to prevent further disease transmission.

The most commonly used diagnostic assays for the detection of *Brucella* infections include serologic tests such as the serum agglutinin test, the Rose Bengal plate test, and ELISA. Molecular methods, including polymerase chain reaction (PCR) and quantitative PCR (qPCR), are gaining ground as diagnostic tests in microbiology, whereas their application in pathogenic typing has provided significant information on pathogen properties and transmission potential [11,12,13]. There is a growing interest in molecular-based detection techniques for diagnosing *Brucella* since these methods can overcome pathogen isolation problems. At the same time, they can also be used efficiently in various biological fluids and pure bacterial cultures [14,15]. Real-time or PCR (qPCR) assays, such as those targeting the bcsp31 gene or the *IS711* insertion sequence, have also been developed to detect Brucella [16] rapidly.

Although PCR has been used to diagnose brucellosis, only a few studies have been conducted to evaluate the use of PCR analysis as a diagnostic tool in field animal samples. This study aimed to assess the infection rate of *Brucella* spp. in ruminants and to evaluate the agreement between a serologic test and a molecular method for the detection of infected cases.

## 2. Materials and Methods

### 2.1. Collection of Samples

All samples were acquired directly from the Laboratory of Brucellosis and Bacterial Disease Serology, Directorate of Athens Veterinary Center, Ministry of Rural Development and Food, Athens, and were kept at −18 °C until the analysis. Samples of whole blood, serum and milk were collected from all 136 ruminants (44 cows, 88 sheep and 4 bulls). None of the animals tested had received any vaccinations, nor was there any herd history of abortion.

The milk samples were collected with an aseptic method from the teats of the udder, which had been cleaned with water and soap, and then their surface was sterilized with 70% ethanol. The first sample of milk was discarded for each animal, and then 15 mL of milk was collected in sterile Falcon tubes. Blood was withdrawn from each animal and was put in two different tubes; the whole blood sample for the PCR analysis was collected in tubes containing ethylene diamine tetra-acetic acid (EDTA), while the serum sample for the serological test was collected in tubes without anticoagulant. The milk and blood samples were immediately transferred to the laboratory and kept at −18 °C until the analysis. 

### 2.2. Serological Method

Serum samples were tested by the Rose Bengal plate test (RBPT) in the Laboratory of Brucellosis. For serology, blood samples were centrifuged (3000× *g* for 10 min), and the serum was divided into aliquots and stored at –20 °C until needed. All sera were evaluated using the Rose Bengal test (MonLab) and the serum agglutination test. The serum sample (30 μL) was transferred to a circle on the white slide, gently but thoroughly mixing the Rose Bengal Brucella antigen [17]. One drop of the reagent was delivered to each serum sample using the dropper provided. Using a clean mixing stick for each specimen, the serum was mixed, and the reagent was spread over the entire circle. The slide was rotated for 4 min. The presence of agglutination was observed under a bright light source.

### 2.3. Positive Control Preparation 

Reference strains of *B. melitensi**s* ATCC 23456 and *B. abortus* ATCC 23448 were purchased from Culture Public Health England (Salisbury, UK), grown at 36.5 °C for 24–36 h on liquid blood culture, and resuspended in phosphate-buffered saline (PBS) buffer to an optical density of 0.5 McFarland units (Densimat Densitometer; Biomerieux Biotechnology, Cheshire, UK). Ten-fold serial dilutions (down to 10^1^ CFU/mL) in PBS were made from 10^7^ CFU/mL bacterial suspensions. Blood and milk samples were spiked by a strain of *B. melitensis* and *B. abortus* with serial ten-fold dilutions to obtain 10^1^–10^3^ CFU/mL of each microorganism. The samples were designated as positive controls to find the detection limit of PCR. 

### 2.4. Genomic DNA Extraction

DNA was directly extracted from blood and milk after centrifugation at 15,000× *g* for 5 min to remove the supernatant from the milk specimens using an automatic extractor (Zybio Company) and following the protocol recommended by the supplier. The DNA of the strains *B. melitensis* and *B. abortus* was extracted with the Nucleic Acid Isolation Kit (Magnetic Beads Method) (Zybio Company, Chongqing, China) following its specific protocol. The purity and the quantity of extracted DNA were evaluated spectrophotometrically by calculating OD260/OD280 (spectrophotometer Epoch, Biotek, London, UK). The DNA was then stored at −20 °C until use.

### 2.5. Identification of Brucella spp. DNA by PCR Amplification

Blood and milk samples were analyzed by PCR for the identification of *Brucella* spp. After the assessment of DNA concentration, PCR was carried out. The B4 and B5 primers described previously [1] were used to detect the *Brucella* genus, which encodes a cell surface protein of *B. abortus* that is 31 kDa, BCSP31. The chosen primers amplify a size of 223 bp for the *Brucella* spp. The specific primers for *B. abortus* and *B. melitensis* were previously described [18] and shown to exploit the multi-copy element *IS711* by using a common primer anchored in the IS element and species-specific primers that bind to the unique sequence flanking the insertion site [19]. *IS711*-PCR is considered highly sensitive and specific for the safe detection of *B. abortus and B. melitensis* [20]. The insertion sequence varies according to the *IS711* copy number. These species-specific primers amplify a size of 498 bp for the *B. abortus* and 731 bp for the *B. melitensis* using 0.2 μM of primers (Table 1) [12,19].

PCR for *Brucella* genes was conducted in a 50 μL final volume solution using Master Mix (PCRBIO TaqMix Red). The amplification was performed by a thermal cycler (96-well thermal cycler, Applied Biosystems, Singapore) as follows: initial denaturation at 95 °C for 180 s; 40 cycles with the following step-cycle profile: denaturation at 95 °C for 15 s; annealing at 60 °C for 15 s; extension at 72 °C for 60 s; and final extension at 72 °C for 600 s. The PCR assays for *IS711* were performed in a total volume of 50 μL that contained the same mix used for *BCSP31*-PCR. The programs for amplification of *B. abortus* and *B. melitensis* consisted of initial DNA denaturation at 95 °C for 180 s and then 35 cycles at 95 °C for 90 s, 65 °C for 60 s, and 72 °C for 60 s. A final extension step of 300 s at 72 °C was performed. PCR products were separated in 2% agarose gel, stained with ethidium bromide (0.5 μg/mL) and documented under UV illumination using the MiniBIS Pro device (DNR Bio-Imaging Systems Ltd., P.O Box 72, Neve Yamin 4492000, Israel).

The specificity of the primers for the detection of *Brucella* spp. DNA was evaluated using a variety of microorganisms that have an antigenic relationship with *Brucella* spp. and may cause false-positive results in serological tests. The absence of DNA amplification in these species validated the primers’ specificity for *Brucella* spp. *DNA.* Moreover, to ensure and evaluate the sensitivity of PCR analysis to detect *Brucella* spp. DNA, at least one negative control sample, one positive control sample from culture, and one positive control sample were tested. The PCR analysis verified all the positive controls, showing that the sensitivity of PCR in spiked blood and milk samples was 10 CFU/mL.

### 2.6. Ethical Approval and Informed Consent

The Institutional Animal Ethics Committee approved the study, and the authors have obtained permission from the farm owners to publish data. Samples were collected in compliance with EU legislation on research involving animals. The farmers were informed about the purpose and methods of the study and that participation was voluntary. All data were handled anonymously, and there was no data collection regarding the identity of individual animals or farmers. 

### 2.7. Statistical Analysis

Statistical analysis included descriptive statistics for the results of the PCR analysis and the serological method. The positive rates between the samples of different types of animals were compared using the chi-squared test. Moreover, the agreement between the results of the two methods was determined using the kappa statistic and the respective *p*-value. To evaluate the observed agreement based on the kappa value, the arbitrary benchmarks for the strength of agreement are described by Thrusfield [21]. Stata 15.0 software was used for analysis (Stata Corp., College Station, TX, USA). For all tests, a *p*-value lower than 0.05 indicates statistical significance.

## 3. Results

The results of the Rose Bengal plate test and PCR analysis in the tested samples are presented in Table 2. The Rose Bengal adhesion response was positive in 12 (8.8%) out of the 136 tested serum samples. Regarding the 12 samples with the positive result on the Rose Bengal plate test (seropositive), 10 samples were from sheep (11.3%; 10 out of 88 sheep samples), and 2 were from cows (4.5%; 2 out of 44 cow samples). The positive rates between the sheep and cow samples were statistically similar (*p* = 0.19). Regarding the results of the PCR analysis, 2 cow samples (1.4%; 2 out of 136 blood and milk tested samples) were positive for both primers for *Brucella* spp. (targeting the gene sequence that encodes the BCSP31 protein) and *B. abortus* (targeting the gene sequence *IS711*) producing 223 and 498 bp amplicons, respectively. Interestingly, the Rose Bengal adhesion response was negative in these 2 cows, and the PCR analysis was positive. The two cows with the positive results on the PCR analysis belong to two different herds in which there were no seropositive animals. Based on kappa statistics, there was only a slight agreement between the PCR analysis and the Rose Bengal plate test results (k= −0.02), and this agreement was not statistically significant (*p* = 0.67). 

## 4. Discussion

According to the World Health Organization, the diagnosis of brucellosis in animals should be made at the farm level. To reduce the incidence of this zoonotic infection among humans, the pathogens must be controlled in the animal population. The disease is most frequently transmitted through unpasteurized dairy products or direct contact of farmers, veterinarians, or laboratory workers with infected animals, tissues or fluids associated with abortion [22,23]. The diagnosis of brucellosis in animals can be achieved by analyzing fluid samples (such as milk and blood samples) [24]. The accurate and early diagnosis of brucellosis is critical for the success of efficient public health measures. 

The present study performed two different diagnostic methods (serological test and PCR analysis). Based on the study results, the infection rate was 8.8% on serological testing and 1.4% on PCR analysis. Interestingly, the positive samples on PCR analysis were negative on serological testing, indicating that in some cases, PCR can detect *Brucella* genes in seronegative animals due to its ability to detect small amounts of the pathogen in body fluids of infected animals [25]. On the other hand, the positive results of serological testing were not confirmed by detecting Brucella DNA on PCR analysis. Therefore, combining the two methods increased the overall sensitivity of laboratory testing for infected cases, providing a positive rate of 10.3%. The positive PCR result in seronegative samples indicates that the positive animals on PCR analysis were probably infected by *B. abortus.* However, they had not developed an immunological reaction capable of producing antibodies that could be detected by the standard serological method. Because the incubation period of the disease varies considerably between animals, such cases of seronegative animals with positive PCR results could be an important source of transmission to other animals or humans [26]. The positive serological results in negative samples on PCR analysis may indicate more chronic cases since antibody levels remain high for a long time, while circulating bacterial DNA is present for a shorter time after the infection, mainly during the septicemic stage

The presence of Brucella spp. is ascertained either indirectly by immunological tests or directly by culture isolation; however, the latter is time-consuming, laborious, and needs a biosafety level 3 laboratory [27]. On the other hand, although serological tests are easy to perform and have the advantage of rapid and sensitive results, they lack specificity due to cross-reactions with other bacteria, particularly with *Yersinia enterocolitica* (O:9, which results from the O chains’ antigenic similarity), *Campylobacter fetus*, *Vibrio cholera*, *Bordetella bronchiseptica* and *Salmonella* spp. [14,27]. Molecular analysis such as PCR has been gaining ground over the past decades and may have a role in monitoring animals for possible infections. PCR results can be especially useful in detecting animals and humans with negative serological tests for brucellosis, allowing rapid identification of infected cases. Seronegative animals that tested positive on PCR analysis could have been exposed to *Brucella* and then converted to seronegative after a certain period of time when the circulating levels of the pathogen have decreased. They can no longer trigger an immune response severe enough to produce antibodies. Alternatively, if samples are taken at an early stage of the infection within the incubation period (i.e., within the first 14 days), the humoral immune response may have not yet induced detectable levels of antibodies on serological testing [26]. However, these animals are very likely to be infectious. Another possible reason for negative serological results in infected cases may be the low count of *Brucella* spp. in the blood, which cannot produce detectable antibody levels. In all these cases, only the detection of the pathogen’s DNA with molecular methods can provide reliable information about the presence of *Brucella* spp. [27,28].

There are mixed results in the literature regarding the sensitivity and specificity of PCR analysis to detect infected cases compared to serological testing. Although some studies have indicated that PCR is a more sensitive and specific method than serological tests for diagnosing *Brucella* in animals, there are several studies demonstrating a higher sensitivity for serological testing [27,29,30,31]. The higher sensitivity of serological testing was also evident in our study since the positive rate for the RBP test was higher than that of PCR analysis. In a human study [32] including 31 blood samples, the reported sensitivity of PCR was 61%, while the sensitivity of serological testing in the respective serum samples was 94%. The authors of this study highlighted that the combined sensitivity of whole blood and serum samples was 97%, similar to our study. The combination of molecular and serological methods increased sensitivity for positive results. In another study [33], Romero et al. tested 93 milk samples (56 culture-positive and 37 negative) using PCR and ELISA to assess the sensitivity and specificity of these two methods. The authors of this study reported that although the specificity was 100% for both methods, the sensitivity was 87.5% (49 samples) for the PCR analysis and 98.2% (55) for ELISA. Wareth et al. [34] examined 215 bovine and bison milk samples from otherwise healthy animals without any signs of infection using ELISA and PCR analysis to detect antibodies against *Brucella* spp. and bacterial genetic material, respectively. Based on their results, there were 34 positive samples (16%) on ELISA and 17 positive samples (7.9%) on PCR analysis. The authors concluded that the lower rate of positive results on PCR analysis than ELISA was because antibody levels remain high for a long time after infection, while circulating bacterial DNA is present for a shorter time. This may be the cause for the higher positive results on RBP test compared to PCR analysis which was found in our study. They noted, however, that some false-positive results on ELISA may be due to cross-reactions from other bacteria (e.g., *Yersinia enterocolitica* O:9). As opposed to the previous studies, Lindahl-Rajala et al. [35] examined 564 bovine milk using ELISA and PCR analysis and reported a higher rate of positive results on PCR analysis (10.3%; 58 out of 564) compared to ELISA (2.12%; 12 out of 564). All seropositive samples were tested positive for the detection of genetic material (PCR), and the authors reported that the molecular assay was more reliable than serological analysis for the detection of infected cases.

There are some limitations of the study that must be addressed. First, the sample size is relatively small to draw conclusions regarding the infection rate or to evaluate other epidemiological indices for Brucellosis in ruminants in Greece. However, even in this small population, our finding of positive PCR results in seronegative animals indicates that PCR is valuable for increasing the overall sensitivity of detection of infected animals. Second, we did not include any gold standard method for the presence of *Brucella* spp. such as cultures to validate the results of the two methods and evaluate the true positive and negative results. Moreover, we did not perform ELISA for additional comparison of laboratory testing since ELISA is the most reliable serological method for diagnosis of Brucellosis, and the possibility of false-positive or negative results on the Rose Bengal plate test cannot be excluded. Lastly, another limitation of this study is that PCR analysis and RBP testing were not performed in the same type of biological fluid for a particular animal (i.e., serum vs. whole blood or milk), which would allow a more direct comparison of the results of the two evaluated methods.

## 5. Conclusions

Brucellosis is an important problem for public health, with a higher incidence in some countries such as Greece, where Brucellosis in cattle, sheep and goats is still endemic. Early and accurate diagnosis of this infection is of great significance for preventing disease transmission and eradicating brucellosis. Although serological tests are more commonly used to diagnose Brucellosis, these tests may have a low sensitivity in the early or latent stages of infection, during which infected animals are asymptomatic. We found that PCR analysis can increase the detection rate of infected animals since samples of two seronegative cows in our study were positive on PCR analysis. This finding indicates that there may be infected animals that have not yet developed an immunological reaction that can be detected by serological tests, highlighting the importance of additional testing to detect possible sources of disease transmission. However, we must note that the serological test is sustainable in terms of cost and time screening tests in herds or flock for Brucella infection. In contrast, PCR analysis cannot be easily applied to many samples because of its cost or because the animals are on a dry period for milk sampling. Moreover, PCR analysis risks false-negative results when the animals are not in a septicemic stage or if they excrete the pathogen at undetectable levels. Therefore, a combination of molecular methods such as PCR with one of the commonly used serological tests to efficiently detect brucellosis, especially in animals of high economic value, could be valuable.

## Figures and Tables

**Table 1 vetsci-09-00193-t001:** Oligonucleotides for the detection of *Brucella* used in this study.

Target	Primers	Oligonucleotide Sequence	Amplified Product
*BCSP31*	B4	5′-TGGCTCGGTTGCCAATATCAA-3′	223 bp
	B5	5′-CGCGCTTGCCTTTCAGGTCTG-3	
*IS711* *B. abortus*	F	5′-TGCCGATCACTTAAGGGCCTTCAT-3′	498 bp
	R	5′-GAC GAACGGAATTTTTCCAATCCC-3′	
*IS711* *B. melitensis*	F	5′-TGCCGATCACTTAAGGGCCTTCAT-3′	731 bp
	R	5′-AAA TCGCGTCCTTGCTGGTCTGA-3′	

**Table 2 vetsci-09-00193-t002:** Results were obtained by Rose Bengal and PCR methods.

Type of Animal	Biological Fluid	Seropositive RB (+)	Seronegative RB (−)	PCR Positive (+)	PCR Negative (−)
Sheep (*n* = 88)	Whole blood	n/a	n/a	0/88	88/88
	Milk	n/a	n/a	0/88	88/88
	Serum	10/88	78/88	n/a	n/a
Cow (*n* = 44)	Whole blood	n/a	n/a	2/44	42/44
	Milk	n/a	n/a	2/44	42/44
	Serum	2/44	42/44	n/a	n/a
Bull (*n* = 4)	Whole blood	n/a	n/a	0/4	4/4
	Serum	0/4	4/4	n/a	n/a

n/a: non-applicable.

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
