# Peer review of "The Importance of Complementary PCR Analysis in Addition to Serological Testing for the Detection of Transmission Sources of *Brucella* spp. in Greek Ruminants"

_vetsci, 2022, doi:10.3390/vetsci9040193_

Round 1
Reviewer 1 Report
Reviewer blind comments to Author
The paper describes a “classic” comparison between serological and Molecular Biology laboratory methods/tools in order to strengthen the importance in the use of combined test for early detection of Brucellosis outbreak. This issue discussed in the paper isn’t really new and also it is a matter of fact that screening test always concern some critical points.
Anyhow it adds more proves of the importance to apply also Molecula Biology "alternative test" to investigate the herd in order to avoid false negatives.
Abstract
Line 23: please, insert a comma after “zoonotic infection”
Line 28: for and not or
Line 26: Which sample were analyzed? In the keywords I read milk, but no mention in the abstract.
Keywords: I suggest to add RBPT
Introduction
Materials and Methods
2.1. Collection of samples: Milk and blood were collected from all animals? Please, specify
Lane 88: authors report that blood samples were collected in tubes containing EDTA. May be this is a mistake ? Why did you use EDTA tubes to colect and analyze serum ? Why didn't you use tubes without anticoagulant ? Later authors report that they also extracted DNA from blood. The type and use of the samples should all be clarified in this section.
2.2 Serological method
Line 95: please, not start the sentence with a number
Please, add a reference for Rose Bengal test
2.3. Spiking experiments I suggest to change the title of this paragraph. Maybe it’s better “Positive control preparation”
Line 104: “non-selective media”… clarify about incubation time (one night is not enough).
2.4. Genomic DNA extraction
Line 112: “after a centrifugation for 5min in order to remove the supernatant”…not is clear ... Did you centrifuge only milk? How many rpm?
2.5. Identification of Brucella spp DNA by PCR amplification
Line 120: “checking for the purity of DNA” Did you mean after assessing DNA concentration ? Please change.
Line 123: correct BCSP31
Line 131: delete this sentence, not is appropriate in material and method section
Please, describe the thermal program always in the same way: seconds or minutes
Results
Authors compare results obtained from serological test and from PCR. But which PCR? In materials and methods it seems that they carried out two different ones.
Furthermore, the comparison between the two different PCR should also be reported: do they give the same results? Are the results different? How significant is the difference, if any?
Comments
Authors should also comment the fact that RBPT positive weren't confirmed by detection of Brucella DNA...
Anyhow we have to remember that RBPT is a sustainable, in term of cost and time, screening test in herds or flock for Brucella infection whereas PCR is a test not easily applied to massive number of samples because of its cost and also because there is the risk that we aren't in a septicemic stage (for blood) or (for milk) cow or ewe are on dry period or they don't necessarily excrete the pathogen in detectable rate to be detected.
I believe these comment must be considered for conclusion.
Author Response
Athens, 29 March 2022
Dear Editors,
We would like to thank you and the reviewers for evaluating our manuscript entitled: “The importance of complementary PCR analysis in addition to serological testing for the detection of transmission sources of Brucella spp. in Greek ruminants”. In the revised version of the manuscript, we tried to address adequately all points raised by the reviewers. Our responses to each point brought up are provided below. (Original comments in bold). Changes in the revised manuscript have been highlighted in red.
First Reviewer
Abstract
Line 23: please, insert a comma after “zoonotic infection”
Authors response: A comma has been inserted.
Line 28: for and not or
Authors response: The word “or” has been corrected to for
Line 26: Which sample were analyzed? In the keywords I read milk, but no mention in the abstract.
Authors response: Both blood and milk samples were analyzed. This is now mentioned in the revised version of the abstract. Also, the word milk has been deleted from the keywords.
Keywords: I suggest to add RBPT
Authors response: The acronym “RBPT” has been added to the keywords.
Materials and Methods
2.1. Collection of samples: Milk and blood were collected from all animals? Please, specify
Authors response: The milk and blood samples were collected from all animals. This is now specified in the subsection of Methods.
Lane 88: authors report that blood samples were collected in tubes containing EDTA. May be this is a mistake ? Why did you use EDTA tubes to colect and analyze serum ? Why didn't you use tubes without anticoagulant ? Later authors report that they also extracted
DNA from blood. The type and use of the samples should all be clarified in this section.
Authors response: We agree with the author that further clarification is needed for the blood samples. Blood was withdrawn from each animal and was put in two different tubes; the whole blood sample for the PCR analysis was collected in tubes containing Ethylene diamine tetra-acetic acid (EDTA), while the serum sample for the serological test was collected in tubes without anticoagulant. This is clarified in the respective part of the manuscript.
2.2 Serological method
Line 95: please, not start the sentence with a number
Authors response: This sentence has been modified so now it does not start with a number.
Please, add a reference for Rose Bengal test
Authors response: A reference to Rose Bengal test has been added
2.3. Spiking experiments I suggest to change the title of this paragraph. Maybe it’s better “Positive control preparation”
Authors response: This title has been changed to “Positive control preparation” according to reviewer’s suggestion
Line 104: “non-selective media”… clarify about incubation time (one night is not enough).
Authors response: The sentence was corrected as: for 24-36h on liquid blood culture
2.4. Genomic DNA extraction
Line 112: “after a centrifugation for 5min in order to remove the supernatant”…not is clear ... Did you centrifuge only milk? How many rpm?
Authors response: It was corrected. Only the milk specimens were centrifugated at 15000xg
2.5. Identification of Brucella spp DNA by PCR amplification
Line 120: “checking for the purity of DNA” Did you mean after assessing DNA concentration ? Please change.
Authors response: This has been changed to “after assessment of DNA concentration”
Line 123: correct BCSP31
Authors response: This has been corrected
Line 131: delete this sentence, not is appropriate in material and method section
Authors response: This sentence has been deleted
Please, describe the thermal program always in the same way: seconds or minutes
Authors response: The thermal program is now described in a consistent way (seconds).
Results
Authors compare results obtained from serological test and from PCR. But which PCR? In materials and methods it seems that they carried out two different ones.
Furthermore, the comparison between the two different PCR should also be reported: do they give the same results? Are the results different? How significant is the difference, if any?
Authors response: Two PCR analyses were performed for each whole blood sample and each milk sample that were collected from each animal. In the first PCR analysis the primers that were used were targeting a gene sequence that is common in all Brucella spp (encoding the protein BCSP31), while in the second PCR analysis the primers were targeting gene sequences that are specific for B. abortus and B. melitensis. The two positive PCR blood and milk samples were positive for the both gene sequence; the one that is common in all Brucella spp (encoding the BSCP31 protein) and the one that is specific for the B. abortus (IS711). This is further clarified in the results section. Specifically we report now that the two samples were positive for both primers for Brucella spp. and B. abortus
Comments
Authors should also comment the fact that RBPT positive weren't confirmed by detection of Brucella DNA...
Authors response: We agree with the reviewer that is has also need to be discussed in the Discussion section. The positive serological results in negative samples on PCR analysis may indicate more chronic cases since antibody levels remain high for a long time, while circulating bacterial DNA is present for a shorter time after the infection, mainly during the septicemic stage. This has been also reported in othe similar studies [1]. The issue of positive RBPT results with negative PCR analysis is now discussed in the second paragraph of the Discussion section.
Anyhow we have to remember that RBPT is a sustainable, in term of cost and time, screening test in herds or flock for Brucella infection whereas PCR is a test not easily applied to massive number of samples because of its cost and also because there is the risk that we aren't in a septicemic stage (for blood) or (for milk) cow or ewe are on dry period or they don't necessarily excrete the pathogen in detectable rate to be detected.
I believe these comment must be considered for conclusion.
Authors response: The points raised by the reviewer about the drawbacks of PCR analysis are now discussed in the conclusion paragraph of the manuscript.
Reviewer 2 Report
The manuscript is well prepared, and the limitations of the study are pointed out.
Line 81 and 90: Check the format of degree symbol
Line 95: Never start a sentence with a number – please adjust.
Line 113: specify which extractor
Table 1: add the primer references in an additional column.
Line 190: Add the point after the sentence.
Line 220: Should there not be a trigger for immune system within a persisting infection? I would expect this effect after an infection = PCR negative. In my opinion a major issue cold be false positives/ negatives in RBT. Could this be the case in the cow samples as they might have B. abortus – the goats B. melitensis.
Line 232: check grammar: “have showed”
Author Response
Reviewer 2
The manuscript is well prepared, and the limitations of the study are pointed out.
Authors response: We thank the reviewer for his kind comments.
Line 81 and 90: Check the format of degree symbol
Authors response: The degree symbol has been corrected.
Line 95: Never start a sentence with a number – please adjust.
Authors response: This sentence has been modified so now it does not start with a number.
Line 113: specify which extractor Table 1: add the primer references in an additional column.
Authors response: The primers has been put in an additional column
Line 190: Add the point after the sentence.
Authors response: A period has been added at the end of the sentence.
Line 220: Should there not be a trigger for immune system within a persisting infection? I would expect this effect after an infection = PCR negative. In my opinion a major issue cold be false positives/ negatives in RBT. Could this be the case in the cow samples as they might have B. abortus – the goats B. melitensis.
We agree with the reviewer that this is a point of discussion. We believe that one of the possible explanations for the discrepancy between the positive PCR results and the negative serological results is that maybe seronegative animals that have been tested positive on PCR analysis could have been exposed to Brucella and then converted to seronegative after a certain period of time when the circulating levels of the pathogen have decreased, and they cannot trigger any more an immune response severe enough to produce antibodies. This is further elaborated in the discussion section. The possibility of false positive and negative results on RBT is now also mentioned in the limitation paragraph of the manuscript.
Line 232: check grammar: “have showed
Author response: This phrase has been amended to “have indicated”